# Gradient-based training of Gaussian Mixture Models for High-Dimensional Streaming Data

## Abstract

We present an approach for efficiently training Gaussian Mixture Models (GMMs) by Stochastic Gradient Descent (SGD) with non-stationary, high-dimensional streaming data. Our training scheme does not require data-driven parameter initialization (e.g., k-means) and has the ability to process high-dimensional samples without numerical problems. Furthermore, the approach allows mini-batch sizes as low as 1, typical for streaming-data settings, and it is possible to react and adapt to changes in data statistics (concept drift/shift) without catastrophic forgetting. Major problems in such streaming-data settings are undesirable local optima during early training phases and numerical instabilities due to high data dimensionalities.We introduce an adaptive annealing procedure to address the first problem,whereas numerical instabilities are eliminated by using an exponential-free approximation to the standard GMM log-likelihood. Experiments on a variety of visual and non-visual benchmarks show that our SGD approach can be trained completely without, for instance, k-means based centroid initialization, and compares to a favorably online variant of Expectation-Maximization (EM) – stochastic EM (sEM), which it outperforms by a large margin for very high-dimensional data.

## 1 Introduction

This contribution focuses Gaussian Mixture Models (GMMs), which represent a probabilistic unsupervised model for clustering and density estimation and allowing sampling and outlier detection. GMMs have been used in a wide range of scenarios, e.g., Melnykov & Maitra (2010). Commonly, free parameters of a GMM are estimated by using the Expectation-Maximizations (EMs) algorithm (Dempster et al., 1977), as it does not require learning rates and automatically enforces all GMM constraints. A popular online variant is stochastic Expectation Maximization (sEM) (Cappé & Moulines, 2009), which can be trained mini-batch wise and is, thus, more suited for large datasets or streaming data.

### 1.1 Motivation

Intrinsically, EM is a batch-type algorithm. Memory requirements can therefore become excessive for large datasets. In addition, streaming-data scenarios require data samples to be processed one by one, which is impossible for a batch-type algorithm. Moreover, data statistics may be subject to changes over time (concept drift/shift), to which the GMM should adapt. In such scenarios, an online, mini-batch type of optimization such as SGD is attractive, as it can process samples one by one, has modest, fixed memory requirements and can adapt to changing data statistics.

### 1.2 Related Work

**Online EM** is a technique for performing EM mini-batch wise, allowing to process large datasets. One branch of previous research Newton et al. (1986); Lange (1995); Chen et al. (2018) has been devoted to the development of stochastic Expectation Maximization (sEM) algorithms that reduce the original EM method in the limit of large batch sizes. The variant of Cappé & Moulines (2009) is widely used due to its simplicity and efficiency for large datasets. These approaches come at the price of additional hyper-parameters (e.g., learning rate or mini-batch size), thus, removing a key advantage of EM over SGD. Another common approach is to modify the EM algorithm itself by, e.g., including heuristics for adding, splitting and merging centroids Vlassis & Likas (2002); Engel

& Heinen (2010); Pinto & Engel (2015); Cederborg et al. (2010); Song & Wang (2005); Kristan et al. (2008); Vijayakumar et al. (2005). This allows GMM-like models to be trained by presenting one sample after another. The models work well in several application scenarios, but their learning dynamics are impossible to analyze mathematically. They also introduce a high number of parameters. Apart from these works, some authors avoid the issue of extensive datasets by determining smaller "core sets" of representative samples and performing vanilla EM Feldman et al. (2011).

**SGD for training GMMs** has, as far as we know, been recently treated only by Hosseini & Sra (2015; 2019). In this body of work, GMM constraint enforcement is ensured by using manifold optimization techniques and re-parameterization/regularization. Thereby, additional hyper-parameters are introduced. The issue of local optima is sidestepped by a k-means type centroid initialization, and the used image datasets are low-dimensional (36 dimensions). Additionally, enforcing positive definiteness constraints by Cholesky decomposition is discussed.

**Annealing and Approximation approaches for GMMs** were proposed by Verbeek et al. (2005); Pinheiro & Bates (1995); Ormoneit & Tresp (1998); Dognin et al. (2009). However, the regularizers proposed by Verbeek et al. (2005); Ormoneit & Tresp (1998) significantly differ from our scheme. GMM log-likelihood approximations, similar to the one used here, are discussed in, e.g., Pinheiro & Bates (1995) and Dognin et al. (2009), but only in combination with EM training.

**GMM Training in High-Dimensional Spaces** is discussed in several publications: A conceptually very interesting procedure is proposed by Ge et al. (2015). It exploits the properties of high-dimensional spaces in order to achieve learning with a number of samples that is polynomial in the number of Gaussian components. This is difficult to apply in streaming settings, since higher-order moments need to be estimated beforehand, and also because the number of samples usually cannot be controlled in practice. Training GMM-like lower-dimensional factor analysis models by SGD on high-dimensional image data is successfully demonstrated in Richardson & Weiss (2018), avoiding numerical issues, but, again, sidestepping the local optima issue by using k-means initialization. The numerical issues associated with log-likelihood computation in high-dimensional spaces are generally mitigated by using the "logsumexp" trick Nielsen & Sun (2016), which is, however, insufficient for ensuring numerical stability for particularly high-dimensional data, such as images.

### 1.3 GOALS AND CONTRIBUTIONS

The goals of this article are to establish GMM training by SGD as a simple and scalable alternative to sEM in streaming scenarios with potentially high-dimensional data. The main novel contributions are:

- a proposal for numerically stable GMM training by SGD that outperforms sEM for high data dimensionalities
- an automatic annealing procedure that ensures SGD convergence from a wide range of initial conditions without prior knowledge of the data (e.g., no k-means initialization) which is especially beneficial for streaming data
- a computationally efficient method for enforcing all GMM constraints in SGD

Apart from these contents, we provide a publicly available TensorFlow implementation.[1]

## 2 DATASETS

We use a variety of different image-based datasets as well as a non-image dataset for evaluation purposes. All datasets are normalized to the $[0, 1]$ range.

**MNIST** (LeCun et al., 1998) contains gray scale images, which depict handwritten digits from 0 to 9 in a resolution of $28 \times 28$ pixels – the common benchmark for computer vision systems.

**SVHN** (Wang et al., 2012) contains color images of house numbers (0-9, resolution $32 \times 32$).

**FashionMNIST** (Xiao et al., 2017) contains gray scale images of 10 cloth categories and is considered as more challenging classification task compared to MNIST.

**Fruits 360** (Murean & Oltean, 2018) consists of colored pictures showing different types of fruits ($100 \times 100 \times 3$ pixels). The ten best-represented classes are selected from this dataset.

**Devanagari** (Acharya et al., 2016) includes gray scale images of handwritten Devanagari letters with a resolution of $32 \times 32$ pixels – the first 10 classes are selected.

**NotMNIST** (Yaroslav Bulatov, 2011) is a gray scale image dataset (resolution $28 \times 28$ pixels) of

---

[1]https://github.com/gmm-iclr21/sgd-gmm

letters from *A* to *J* extracted from different public available fonts.

**ISOLET** (Cole & Fanty, 1990) is a non-image dataset containing 7 797 samples of spoken letters recorded from 150 subjects. Each sample was encoded and is represented by 617 float values.

## 3 GAUSSIAN MIXTURE MODELS

GMMs are probabilistic models that intend to explain the observed data $X = \{\boldsymbol{x}_n\}$ by expressing their density as a weighted mixture of $K$ Gaussian component densities $\mathcal{N}(\boldsymbol{x}; \boldsymbol{\mu}_k, \boldsymbol{P}_k) \equiv \mathcal{N}_k(\boldsymbol{x})$: $p(\boldsymbol{x}) = \sum_k^K \pi_k \mathcal{N}_k(\boldsymbol{x})$. We work with precision matrices $\boldsymbol{P} = \boldsymbol{\Sigma}^{-1}$ instead of covariances $\boldsymbol{\Sigma}$. This is realized by optimizing the (incomplete) log-likelihood

$$\mathcal{L} = \mathbb{E}_n \left[ \log \sum_k \pi_k \mathcal{N}_k(\boldsymbol{x}_n) \right]. \tag{1}$$

### 3.1 GMMS AND SGD

GMMs require the mixture weights to be normalized: $\sum_k \pi_k = 1$ and the precision matrices to be positive definite: $\boldsymbol{x}^T \boldsymbol{P}_k \boldsymbol{x} \geq 0 \; \forall \boldsymbol{x}$. These constraints must be explicitly enforced after each SGD step:

**Weights** $\pi_k$ are adapted according to Hosseini & Sra (2015), which replaces them by other free parameters $\xi_k$ from which the $\pi_k$ are computed so that normalization is ensured:

$$\pi_k = \frac{\exp(\xi_k)}{\sum_j \exp(\xi_j)}. \tag{2}$$

**Precision Matrices** need to be positive-definite, so we re-parameterize these as $\boldsymbol{P}_k = (\boldsymbol{D}_k^T \boldsymbol{D}_k)$, where the upper-diagonal matrices $\boldsymbol{D}_k$ result from a Cholesky decomposition. Consequently, $\det \boldsymbol{\Sigma}_k = \det \boldsymbol{P}_k^{-1} = (\det(\boldsymbol{D}_k^T \boldsymbol{D}_k))^{-1} = (\text{tr}(\boldsymbol{D}_k))^{-2}$ can be computed efficiently. To avoid re-computing the costly Cholesky decomposition of the $\boldsymbol{P}_k$ at every iteration, we perform it on the initial precision matrices and just erase the elements below the diagonal in the $\boldsymbol{D}_k$ after each gradient step.

### 3.2 MAX-COMPONENT APPROXIMATION FOR GMMS

The log-likelihood Eq. (1) is difficult to optimize by SGD (see Sec. 3.3). This is why we intend to find a lower bound that we can optimize instead. A simple scheme is given by

$$\mathcal{L} = \mathbb{E}_n \left[ \log \sum_k \pi_k \mathcal{N}_k(\boldsymbol{x}_n) \right] \geq \mathbb{E}_n \left[ \log \max_k (\pi_k \mathcal{N}_k(\boldsymbol{x}_n)) \right] = \hat{\mathcal{L}} = \mathbb{E}_n \left[ \log (\pi_{k^*} \mathcal{N}_{k^*}(\boldsymbol{x}_n)) \right] \tag{3}$$

where $k^* = \arg\max_k \pi_k \mathcal{N}_k(\boldsymbol{x}_n)$. This is what we call the *max-component approximation* of Eq. (3). In contrast to the lower bound that is constructed for EM-type algorithms, this bound is usually not tight. The advantages of $\hat{\mathcal{L}}$ are the avoidance of local optima in SGD, and the elimination of exponentials causing numerical instabilities for high data dimensions. The "logsumexp" trick is normally employed with GMMs to rectify this by factoring out the largest component probability $\mathcal{N}_{k^*}$. This mitigates, but does not avoid numerical problems when distances are high. To give an example: we normalize the component probability $\mathcal{N}_k = e^{-101}$ (using 32-bit floats) by the highest probability $\mathcal{N}_{k^*} = e^3$, and we obtain $\frac{\mathcal{N}_k}{\mathcal{N}_{k^*}} = e^{-104}$, which is numerically problematic.

### 3.3 UNDESIRABLE LOCAL OPTIMA IN SGD TRAINING

An issue when performing SGD without k-means initialization concerns undesirable local optima. **Degenerate Solutions** occur when naively optimizing $\mathcal{L}$ by SGD (see Fig. 1a). All components have the same weight, centroid and covariance matrix: $\pi_k \approx \frac{1}{K}$, $\boldsymbol{\mu}_k = \mathbb{E}[\boldsymbol{X}]$, $\boldsymbol{\Sigma}_k = \text{Cov}(\boldsymbol{X}) \; \forall k$, in which case all gradients vanish (see App. A.3 for a proof). These solutions are avoided by $\hat{\mathcal{L}}$, since only a subset of components is updated by SGD, thereby breaking the symmetry between components.

**Single/Sparse-Component Solutions** occur when optimizing $\hat{\mathcal{L}}$ by SGD (see Fig. 1b). They are characterized by one or several components $\{k_i\}$ that have large weights with centroid and precision

matrices given by the mean and covariance of a significant subset $\boldsymbol{X}_{k_i} \subset \boldsymbol{X}$ of the data $\boldsymbol{X}$: $\pi_{k_i} \gg 0$, $\boldsymbol{\mu}_{k_i} = \mathbb{E}[\boldsymbol{X}_{k_i}]$, $\boldsymbol{\Sigma}_{k_i} = \mathrm{Cov}(\boldsymbol{X}_{k_i})$, whereas the remaining components $k$ are characterized by $\pi_k \approx 0$, $\boldsymbol{\mu}_k = \boldsymbol{\mu}(t\!=\!0)$, $\boldsymbol{P}_k = \boldsymbol{P}(t\!=\!0)$. Thus, these unconverged components are almost never best-matching components $k^*$. The max-operation in $\hat{\mathcal{L}}$ causes gradients like $\frac{\partial \hat{\mathcal{L}}}{\partial \boldsymbol{\mu}_k}$ to contain $\delta_{kk^*}$ (see App. A.3). This implies that they are non-zero only for the best-matching component $k^*$. Thus the gradients of unconverged components vanish, implying that they remain in their unconverged state.


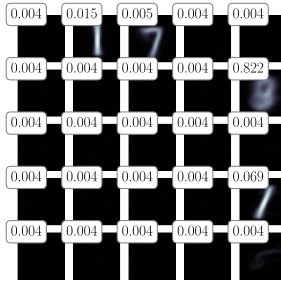

a) degenerate solution (optimizing $\mathcal{L}$)  b) sparse-component solution (optimizing $\hat{\mathcal{L}}$)

Figure 1: Undesirable solutions during SGD, visualized for MNIST with component weights $\pi_k$.

### 3.4 ANNEALING PROCEDURE FOR AVOIDING LOCAL OPTIMA

Our approach for avoiding undesirable solutions is to punish their characteristic response patterns by a modification of $\hat{\mathcal{L}}$, the *smoothed max-component log-likelihood* $\hat{\mathcal{L}}^\sigma$:

$$\hat{\mathcal{L}}^\sigma = \mathbb{E}_n \mathrm{max}_k \left[ \sum_j \boldsymbol{g}_{kj}(\sigma) \log \left( \pi_j \mathcal{N}_j(\boldsymbol{x}_n) \right) \right] = \mathbb{E}_n \sum_j \boldsymbol{g}_{k^*j}(\sigma) \log \left( \pi_j \mathcal{N}_j(\boldsymbol{x}_n) \right). \quad (4)$$

The entries of the $\boldsymbol{g}_k$ are computed by a Gaussian function centered on component $k$ with common spatial standard deviation $\sigma$, where we assume that the $K$ components are arranged on a $\sqrt{K} \times \sqrt{K}$ grid with 2D Euclidean metric (see App. A.4). Eq. (4) essentially represents a smoothing of the $\log (\pi_k \mathcal{N}_k(\boldsymbol{x}))$ with a 2D convolution filter (we use periodic boundary conditions). Thus, Eq. (4) is maximized if the log probabilities follow a uni-modal Gaussian profile of spatial variance $\sim \sigma^2$, which heavily punishes single-component solutions that have a locally delta-like response.

Annealing starts with large values of $\sigma(t) = \sigma_0$ and reduces it over time to an asymptotic small value of $\sigma = \sigma_\infty$, thus, smoothly transitioning from $\hat{\mathcal{L}}^\sigma$ in Eq. (4) into $\hat{\mathcal{L}}$ in Eq. (3).

**Annealing Control** is ensured by adjusting $\sigma$, which defines an effective upper bound on $\hat{\mathcal{L}}^\sigma$ (see App. A.2 for a proof). This implies that the loss will be stationary once this bound is reached, which we consider a suitable indicator for reducing $\sigma$. We implement an annealing control that sets $\sigma \leftarrow 0.9\sigma$ whenever the loss is considered sufficiently stationary. Stationarity is detected by maintaining an exponentially smoothed average $\ell(t) = (1 - \alpha)\ell(t - 1) + \alpha \hat{\mathcal{L}}^\sigma(t)$ on time scale $\alpha$. Every $\frac{1}{\alpha}$ iterations, we compute the fractional increase of $\hat{\mathcal{L}}^\sigma$ as

$$\Delta = \frac{\ell(t) - \ell(t - \alpha^{-1})}{\ell(t - \alpha^{-1}) - \hat{\mathcal{L}}^\sigma(t = 0)} \quad (5)$$

and consider the loss stationary iff $\Delta < \delta$ (the latter being a free parameter). The choice of the time constant for smoothing $\hat{\mathcal{L}}^\sigma$ is outlined in the following section.

### 3.5 TRAINING PROCEDURE FOR SGD

Training GMMs with SGD is performed by maximizing the smoothed max-component log-likelihood $\hat{\mathcal{L}}^\sigma$ from Eq. (4). At the same time, we enforce the constraints on the component weights and covariances as described in Sec. 3.1 and transition from $\hat{\mathcal{L}}^\sigma$ into $\hat{\mathcal{L}}$ by annealing (see Sec. 3.4). SGD requires a learning rate $\epsilon$ to be set, which in turn determines the parameter $\alpha$ (see Sec. 3.4) as $\alpha = \epsilon$ since stationarity detection should operate on a time scale similar to that of SGD. Cholesky matrices

$\boldsymbol{D}_k$ are initialized to $D_{\max}I$ and are clipped after each iteration so that diagonal entries are in the range $[0, D_{\max}^2]$. This is necessary to avoid excessive growth of precisions for data entries with vanishing variance, e.g., pixels that are always black. Weights are uniformly initialized to $\pi^i = \frac{1}{K}$, centroids in the range $[-\mu^i, +\mu^i]$ (see Alg. 1 for a summary). Please note that our SGD approach requires no centroid initialization by k-means, as it is usually recommended when training GMMs with EM. We discuss and summarize good practices for choosing hyper-parameters Sec. 5.

---

Algorithm 1: Steps of SGD-GMM training.

---

**Data:** initializer values: $\mu^i$, $K$, $\epsilon_0/\epsilon_\infty$, $\sigma_0/\sigma_\infty$, $\delta$ and data $\boldsymbol{X}$
**Result:** trained GMM model

1  $\boldsymbol{\mu} \leftarrow \mathcal{U}(-\mu^i, +\mu^i),\;\; \pi \leftarrow 1/K,\;\; \boldsymbol{P} \leftarrow ID_{\max},\;\; \sigma \leftarrow \sigma_0,\;\; \epsilon \leftarrow \epsilon_0$      // initialization
2  **forall** $t < T$ **do**      // training loop
3      $\boldsymbol{g}(t) \leftarrow \text{create\_annealing\_mask}(\sigma, t)$      // see Sec. 3.4 and App. A.4
4      $\boldsymbol{\mu}(t), \boldsymbol{P}(t), \pi(t) = \epsilon\frac{\partial\hat{\mathcal{L}}^\sigma}{\partial\boldsymbol{\mu}} + \boldsymbol{\mu}(t\text{-}1), \epsilon\frac{\partial\hat{\mathcal{L}}^\sigma}{\partial\boldsymbol{P}} + \boldsymbol{P}(t\text{-}1), \epsilon\frac{\partial\hat{\mathcal{L}}^\sigma}{\partial\pi} + \pi(t\text{-}1)$      // SGD update
5      $\boldsymbol{P}(t) \leftarrow \text{precisions\_clipping}(\boldsymbol{P}, D_{\max})$      // see Sec. 3.5
6      $\pi(t) \leftarrow \text{normalization}(\pi(t))$      // see Eq. (2)
7      $\ell(t) \leftarrow (1-\alpha)\ell(t\text{-}1) + \alpha\hat{\mathcal{L}}^\sigma(\boldsymbol{x}(t))$      // sliding log-likelihood
8      **if** *annealing update iteration* **then**      // see Sec. 3.4
9        **if** $\Delta < \delta$ **then** $\sigma(t) \leftarrow 0.9\sigma(t-1),\;\; \epsilon(t) \leftarrow 0.9\epsilon(t-1)$      // $\Delta$ see Eq. (5)

---

### 3.6 TRAINING PROCEDURE FOR STOCHASTIC EXPECTATION MAXIMIZATION

We use sEM as proposed by Cappé & Moulines (2009). We choose the step size of the form $\rho_t = \rho_0(t + 1)^{-0.5+\alpha}$, with $\alpha \in [0, 0.5]$, $\rho_0 < 1$ and enforce $\rho(t) \geq \rho_\infty$. Values for these parameters are determined via a grid search in the ranges $\rho_0 \in \{0.01, 0.05, 0.1\}$, $\alpha \in \{0.01, 0.25, 0.5\}$ and $\rho_\infty \in \{0.01, 0.001, 0.0001\}$. Each sEM iteration uses a batch size $B$. Initial accumulation of sufficient statics is conducted for $10\%$ of an epoch, but not when re-training with new data statistics. Parameter initialization and clipping of precisions is performed just as for SGD, see Sec. 3.5.

### 3.7 COMPARING SGD AND SEM

Since sEM optimizes the log-likelihood $\mathcal{L}$, whereas SGD optimizes the annealed approximation $\hat{\mathcal{L}}^\sigma$, the comparison of these measures should be considered carefully. We claim that the comparison is fair assuming that **i)** SGD annealing has converged and **ii)** GMM responsibilities are sharply peaked so that a single component has responsibility of $\approx 1$. It follows from **i)** that $\hat{\mathcal{L}}^\sigma \approx \hat{\mathcal{L}}$ and **ii)** implies that $\hat{\mathcal{L}} \approx \mathcal{L}$. Condition **ii)** is usually satisfied to high precision especially for high-dimensional inputs: if it is not, the comparison is biased in favor of sEM, since $\mathcal{L} > \hat{\mathcal{L}}$ by definition.

## 4 EXPERIMENTS

Unless stated otherwise, the experiments in this section will be conducted with the following parameter values for sEM and SGD (where applicable): mini-batch size $B = 1$, $K = 8 \times 8$, $\mu^i = 0.1$, $\sigma_0 = 2$, $\sigma_\infty = 0.01$, $\epsilon = 0.001$, $D_{\max} = 20$. Each experiment is repeated 10 times with identical parameters but different random seeds for parameter initialization. See Sec. 5 for a justification of these choices. Due to input dimensionality, all precision matrices must be taken to be diagonal. Training/test data are taken from the datasets mentioned in Sec. 2.

### 4.1 ROBUSTNESS OF SGD TO INITIAL CONDITIONS

Here, we train GMMs for three epochs on classes 1 to 9 for each dataset. We use different random and non-random initializations of the centroids and compare the final log-likelihood values. Random centroid initializations are parameterized by $\mu^i \in \{0.1, 0.3, 0.5\}$, whereas non-random initializations are defined by centroids from a previous training run on class 0 (one epoch). The latter is done to have a non-random centroid initialization that is as dissimilar as possible from the training data. The initialization of the precisions cannot be varied because empirical data shows that training converges

to undesirable solutions if the precisions are not initialized to large values. While this will have to be investigated further, we find that convergence to near-identical levels, regardless of centroid initialization for all datasets (see Tab. 1 for more details).

## 4.2 ADDED VALUE OF ANNEALING

To demonstrate the beneficial effects of annealing, we perform experiments on all datasets with annealing turned off. This is achieved by setting $\sigma_0 = \sigma_\infty$. This invariably produces sparse-component solutions with strongly inferior log-likelihoods after training, please refer to Tab. 1.

Table 1: Effect of different random and non-random centroid initializations of SGD training. Given are the means and standard deviations of final log-likelihoods (10 repetitions per experiment). To show the added value of annealing, the right-most column indicates the final log-likelihoods when annealing is turned off. This value should be co,pared to the leftmost entry in each row where annealing is turned on. Standard deviations in this case where very small so they are omitted.

| Initialization | random | | | | | | non-random | | no ann. |
| | $\mu^i = 0.1$ | | $\mu^i = 0.3$ | | $\mu^i = 0.5$ | | init class 0 | | $\mu^i = 0.1$ |
| Dataset | mean | std | mean | std | mean | std | mean | std | mean |
|---|---|---|---|---|---|---|---|---|---|
| MNIST | 205.47 | 1.08 | 205.46 | 0.77 | 205.68 | 0.78 | 205.37 | 0.68 | 124.1 |
| FashionMNIST | 231.22 | 1.53 | 231.58 | 2.84 | 231.00 | 1.11 | 229.59 | 0.59 | 183.0 |
| NotMNIST | $-48.41$ | 1.77 | $-48.59$ | 1.56 | $-48.32$ | 1.13 | $-49.37$ | 2.32 | -203.8 |
| Devanagari | $-15.95$ | 1.59 | $-15.76$ | 1.34 | $-17.01$ | 1.11 | $-22.07$ | 4.59 | -263.4 |
| Fruits 360 | 12 095.80 | 98.02 | 12 000.70 | 127.00 | 12 036.25 | 122.06 | 10 912.79 | 1 727.61 | 331.2 |
| SVHN | 1 328.06 | 0.94 | 1 327.99 | 1.59 | 1 328.40 | 1.17 | 1 327.80 | 0.94 | 863.2 |
| ISOLET | 354.34 | 0.04 | 354.36 | 0.04 | 354.36 | 0.04 | 354.20 | 0.05 | 201.5 |

## 4.3 CLUSTERING PERFORMANCE EVALUATION

To compare the clustering performance of sEM and GMM the Davies-Bouldin score (Davies & Bouldin, 1979) and the Dunn index (Dunn, 1973) are determined. We evaluate the grid-search results to find the best parameter setup for each metric for comparison. sEM is initialized by k-means to show that our approach does not depend on parameter initialization. Tab. 2 indicaties that SGD can egalize sEM performance (see also App. A.5).

Table 2: Clustering performance comparison of SGD and sEM training using Davies-Bouldin score (less is better) and Dunn index (more is better). Results are in bold face whenever they are better by more than half a standard deviation.

| Metric / Algo. / Dataset | Davies-Bouldin score | | | | Dunn index | | | |
| | SGD | | sEM | | SGD | | sEM | |
| | mean | std | mean | std | mean | std | mean | std |
|---|---|---|---|---|---|---|---|---|
| MNIST | 2.50 | 0.04 | **2.47** | 0.04 | 0.18 | 0.02 | 0.16 | 0.02 |
| FashionMNIST | **2.06** | 0.05 | 2.20 | 0.04 | 0.20 | 0.03 | 0.19 | 0.02 |
| NotMNIST | 2.30 | 0.03 | **2.12** | 0.03 | 0.15 | 0.03 | 0.14 | 0.04 |
| Devanagari | **2.60** | 0.04 | 2.64 | 0.02 | **0.33** | 0.01 | 0.27 | 0.04 |
| SVHN | **2.34** | 0.04 | 2.41 | 0.03 | 0.15 | 0.02 | 0.15 | 0.02 |

## 4.4 STREAMING-DATA EXPERIMENTS WITH CONSTANT DATA STATISTICS

We train GMMs for three epochs (enough for convergence in all cases) using SGD and sEM on all datasets as described in Secs. 3.5 and 3.6.The resulting centroids of our SGD-based approach are shown in Fig. 2, whereas the final loss values for SGD and sEM are compared in Tab. 3. The centroids for both approaches are visually similar, except for the topological organization due to annealing for SGD, and the fact that in most experiments, some components do not converge for sEM while the others do. Tab. 3 indicates that SGD achieves performances superior to sEM in the majority of cases, in particular for the highest-dimensional datasets (SVHN: 3 072 dimensions and Fruits 360: 30 000 dimensions).

Table 3: Comparison of SGD and sEM training on all datasets in a streaming-data scenario. Each time mean log-likelihoods (10 repetitions) at the end of training, and their standard deviations are presented. Results are in bold face whenever they are higher by more than half a standard deviation. Additionally, the averaged maximum responsibilities ($p_{k*}$) for test data are given for justifying the max-component approximation.

| Algorithm
Dataset | $\varnothing \max p_{k*}$ | SGD | | sEM | |
|---|---|---|---|---|---|
| | | *mean* | *std* | *mean* | *std* |
| MNIST | 0.992 674 | 216.60 | 0.31 | 216.82 | 1.38 |
| FashionMNIST | 0.997 609 | **234.59** | 2.28 | 222.95 | 6.03 |
| NotMNIST | 0.998 713 | **−34.76** | 1.16 | −40.09 | 8.90 |
| Devanagari | 0.999 253 | **−14.65** | 1.09 | −13.46 | 6.16 |
| Fruits 360 | 0.999 746 | **11754.32** | 75.63 | 5 483.00 | 1 201.60 |
| SVHN | 0.998 148 | **1329.83** | 0.80 | 1 176.07 | 16.91 |
| ISOLET | 0.994 069 | 354.21 | 0.01 | 354.55 | 0.37 |

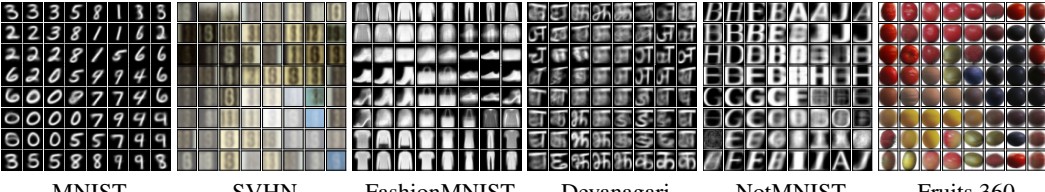

MNIST    SVHN    FashionMNIST    Devanagari    NotMNIST    Fruits 360

Figure 2: Exemplary results for centroids learned by SGD, trained on full images.

**Visualization of High-dimensional sEM Outcomes**   Fig. 3 was obtained after training GMMs by sEM on both the Fruits 360 and the SVHN dataset. It should be compared to Fig. 2, where an identical procedure was used to visualize centroids of SGD-trained GMMs. It is notable that the effect of unconverged components does not occur at all for our SGD approach, which is due to the annealing mechanism that "drags" unconverged components along.

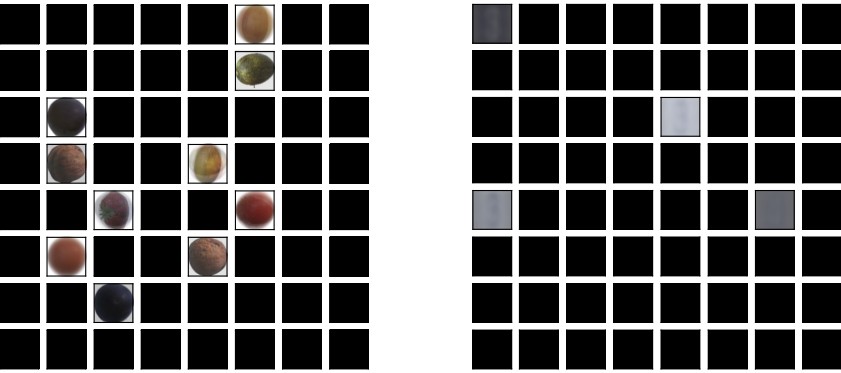

Figure 3: Visualization of centroids after exemplary training runs (3 epochs) on high-dimensional datasets for sEM: Fruits 360 (left, 30 000 dimensions) and SVHN (right, 3 000 dimensions). Component entries are displayed "as is", meaning that low brightness means low RGB values. Visibly, many GMM components remain unconverged, which is analogous to a sparse-component solution and explains the low log-likelihood values especially for these high-dimensional datasets.

## 5   DISCUSSION AND CONCLUSION

The **Relevance of this Article** is outlined by the fact that training GMMs by SGD was recently investigated in the community by Hosseini & Sra (2015; 2019). We go beyond, since our approach does not rely on off-line data-driven model initialization, and works for high-dimensional streaming data.The presented SGD scheme is simple and very robust to initial conditions due to the proposed annealing procedure, see Sec. 4.1 and Sec. 4.2. In addition, our SGD approach compares favorably to the reference model for online EM (Cappé & Moulines, 2009) in terms of achieved log-likelihoods, which was verified on multiple real-world datasets. Superior SGD performance is observed for the high-dimensional datasets.

**Analysis of Results** suggests that SGD performs better than sEM on average, see Sec. 4.4, although the differences are very modest. It should be stated clearly that it cannot be expected, and is not the goal of this article, to outperfom sEM by SGD in the general case, only to achieve a similar performance. However, if sEM is used without, e.g., k-means initialization, components may not converge (see Fig. 3 for a visual impression) for very high-dimensional data like Fruits 360 and SVHN datasets, which is why SGD outperforms sEM in this case. Another important advantage of SGD over sEM is the fact that no grid search for finding certain hyper-parameter values is necessary, whereas sEM has a complex and unintuitive dependency on $\rho_0$, $\rho_\infty$ and $\alpha_0$.

**Small Batch Sizes and Streaming Data** are possible with the SGD-based approach. Throughout the experiments, we used a batch size of 1, which allows streaming-data processing without the need to store any samples at all. Larger batch sizes are, of course, possible and increase execution speed. In the experiments conducted here, SGD (and sEM) usually converged within the first two epochs, which is a substantial advantage whenever huge sets of data have to be processed.

**No Assumptions About Data Generation** are made by SGD in contrast to the EM and sEM algorithms. The latter guarantee that the loss will not decrease due to an M-step. This, however, assumes a non-trivial dependency of the data on an unobservable latent variable (see App. A.1 for a proof). In contrast, SGD makes no such hard-to-verify assumptions, which is a rather philosophical point, but may be an advantage in certain situations where data are strongly non-Gaussian.

**Numerical Stability** is assured by our SGD training approach. It does not optimize the log-likelihood but its max-component approximation. This approximation contains no exponentials at all and is very well justified by the results of Tab. 3 which show that component probabilities are very strongly peaked. In fact, it is the gradient computations where numerical problems (e.g., NaN values) occurred. The "logsumexp" trick mitigates the problem, but does not eliminate it (see Sec. 3.2). It cannot be used when gradients are computed automatically (what most machine learning frameworks do).

**Hyper-Parameter Selection Guidelines** are as follows: the learning rate $\epsilon$ must be set by cross-validation (a good value is 0.001). We empirically found that initializing precisions to the cut-off value $D_{\max}$ and an uniform initialization of the $\pi_i$ are beneficial, and that centroids are best initialized to small random values. A value of $D_{\max} = 20$ always worked in the experiments. Generally, the cut-off must be much larger than the inverse of the data variance. In many cases, it should be possible to estimate this roughly, even in streaming settings, especially when samples are normalized. For density estimation, choosing higher values for $K$ leads to higher final log-likelihoods (validated in App. A.6). For clustering, $K$ should be selected using standard techniques for GMMs. The parameter $\delta$ controls loss stationarity detection for the annealing procedure and was shown to perform well for $\delta = 0.05$. Larger values will lead to faster decrease of $\sigma(t)$, which may impair convergence. Smaller values are always admissible but lead to longer convergence times. The annealing time constant $\alpha$ should be set to the GMM learning rate $\epsilon$ or lower. Smaller values of $\alpha$ lead to longer convergence times since $\sigma(t)$ will be updated less often. The initial value $\sigma_0$ needs to be large in order to enforce convergence for all components. A typical value is $0.25\sqrt{K}$. The lower bound on $\sigma$, $\sigma_\infty$ should be as small as possible to achieve high log-likelihoods (e.g., 0.01, see proof in App. A.2).

## 6 OUTLOOK

The presented work can be extended in several ways: First of all, annealing control could be simplified further by inferring good $\delta$ values from $\alpha$. Likewise, *increases* of $\sigma$ might be performed automatically when the loss rises sharply, indicating a task boundary. As we found that GMM convergence times grow linear with the number of components, we will investigate hierarchical GMM models that operate like a Convolutional Neural Network (CNN), and in which individual GMMs only see a local patch of the input and can therefore have low $K$. Lastly, we will investigate a replay by SGD-trained GMMs for continual learning architectures. GMMs could compare favorably to Generative Adversarial Nets (Goodfellow et al., 2014) due to faster training and the fact that sample generation capacity can be monitored via the log-likelihood.

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

# A SUPPLEMENTARY MATERIAL

## A.1 ASSUMPTIONS MADE BY EM AND SGD

The EM algorithm assumes that the observed data samples $\{\boldsymbol{x}_n\}$ depend on unobserved latent variables $\boldsymbol{z}_n$ in a non-trivial fashion. This assumption is formalized for a GMM with K components by formulating the complete-data likelihood in which $\boldsymbol{z}_n \equiv z_n \in \{0, \ldots, K-1\}$ is a scalar:

$$p(\boldsymbol{x}_n, z_n) = \pi_{z_n} \mathcal{N}_{z_n}(\boldsymbol{x}_n) \tag{6}$$

where we have defined $\mathcal{N}_k(\boldsymbol{x}_n) = \mathcal{N}(\boldsymbol{x}_n; \boldsymbol{\theta}_k, \boldsymbol{\mu}_k)$ for brevity. It is assumed that the $z_n$ are unobservable random variables whose distribution is unknown. Marginalizing them out gives us the incomplete-data likelihood $p(\boldsymbol{x}_n) = \sum_k p(\boldsymbol{x}_n, z_n)$. The derivation of the EM algorithm starts out with the total incomplete-data log-likelihood

$$\begin{aligned}
\mathcal{L} = \log p(X) = \log \prod_n p(\boldsymbol{x}_n) &= \sum_n \log p(\boldsymbol{x}_n) \\
&= \sum_n \log \sum_k p(\boldsymbol{x}_n, z_n = k) \\
&= \sum_n \log \sum_k p(z_n = k) \frac{p(\boldsymbol{x}_n, z_n = k)}{p(z_n = k)}.
\end{aligned} \tag{7}$$

Due to the assumption that $\mathcal{L}$ is obtained by marginalizing out the latent variables, an explicit dependency on $z_n$ can be re-introduced. For the last expression, Jensen' inequality can be used to construct a lower bound:

$$\begin{aligned}
\mathcal{L} =\sim \sum_n \log \sum_k p(\boldsymbol{z}_n = k) \frac{p(\boldsymbol{x}_n, \boldsymbol{z}_n = k)}{p(\boldsymbol{z}_n = k)} \\
\geq \sum_n \sum_k p(\boldsymbol{z}_n = k) \log \frac{p(\boldsymbol{x}_n, \boldsymbol{z}_n = k)}{p(\boldsymbol{z}_n = k)}.
\end{aligned} \tag{8}$$

Since the realizations of the latent variables are unknown, we can assume any form for their distribution. In particular, for the choice $p(z_n) \sim p(\boldsymbol{x}_n, z_n)$, the lower bound becomes tight. Simple algebra and the fact that the distribution $p(z_n)$ must be normalized gives us:

$$\begin{aligned}
p(z_n = k) &= \frac{p(z_n = k, \boldsymbol{x}_n)}{p(\boldsymbol{x}_n)} \\
&= p(z_n = k|\boldsymbol{x}_n) \\
&= \frac{p(z_n = k, \boldsymbol{x}_n)}{\sum_l p(z_n = l, \boldsymbol{x}_n)} \\
&= \frac{\pi_k \mathcal{N}_k(\boldsymbol{x}_n)}{\sum_l \pi_l \mathcal{N}_l(\boldsymbol{x}_n)}
\end{aligned} \tag{9}$$

where we have used Eq. (6) in the last step. $p(z_n = k|\boldsymbol{x}_n)$ is a quantity that can be computed from data with no reference to the latent variables. For GMMs it is usually termed *responsibility* and we write it as $p(z_n = k|\boldsymbol{x}_n) \equiv \gamma_{nk}$.

However, the construction of a tight lower bound, which is actually different from $\mathcal{L}$, only works when $p(\boldsymbol{x}_n, z_n)$ depends non-trivially on the latent variable $z_n$. If this is not the case, we have $p(\boldsymbol{x}_n, z_n) = K^{-1} p(\boldsymbol{x}_n)$ and the derivation of Eq. (8) goes down very differently:

$$\begin{aligned}
\mathcal{L} \sim \sum_n \log p(\boldsymbol{x}_n) &\geq \sum_n \sum_k p(z_n = k) \log \frac{p(\boldsymbol{x}_n, z_n = k)}{p(z_n = k)} \\
&= \sum_n \sum_k p(z_n = k) \log \frac{K^{-1} p(\boldsymbol{x}_n)}{p(z_n = k)} \\
&= \sum_n \log \left( K^{-1} p(\boldsymbol{x}_n) \right) - \sum_k p(z_n = k) \log p(z_n = k) \\
&\equiv \sum_n \left( \log p(\boldsymbol{x}_n) - \left( \log K - \mathcal{H}[z_n] \right) \right)
\end{aligned} \tag{10}$$

where $\mathcal{H}$ represents the Shannon entropy of $p(\boldsymbol{z})$. The highest value this can have is $\log K$ for an uniform distribution of the $z_n$, finally leading to a lower bound for $\mathcal{L}$ of

$$\mathcal{L} \geq \sum_n \Big( \log p(\boldsymbol{x}_n) \Big) = \mathcal{L} \tag{11}$$

which is indeed tight, but trivial, and thus does not simplify the problem at all. In particular, no closed-form solutions to the associated extreme value problem can be computed for this case.

This shows that optimizing GMMs by Expectation-Maximization assumes that each sample has been drawn from a single element in a set of $K$ uni-modal Gaussian distributions. Which distribution is selected for sampling depends on a latent random variable. On the other hand, optimization by SGD uses the incomplete-data log-likelihood $\mathcal{L}$ as basis for optimization, without probabilistic interpretation. This may be advantageous for problems where the assumption of Gaussianity is badly violated, although empirical studies indicate that optimization by EM works very well in a wide range of scenarios.

## A.2 Proof that $\sigma$ Defines an Upper Bound on $\hat{\mathcal{L}}^\sigma$

Let us assume that SGD optimization has reached a stationary point where the derivative w.r.t. all GMM parameters is $0$. In this situation, we claim that the only way to increase the loss is by manipulating $\sigma$. We show here that $\frac{\partial \hat{\mathcal{L}}^\sigma}{\partial \sigma} < 0 \forall \sigma > 0$, and that $\frac{\partial \hat{\mathcal{L}}^\sigma}{\partial \sigma} = 0$ for $\sigma = 0$. This means that the loss can be increased by decreasing $\sigma$, up to the point where $\sigma = 0$.

For each sample, the 2D profile of $\log(\pi_k \mathcal{N}_k) \equiv f_k$ is assumed to be radially symmetric, centered on the best-matching component $k^*$ and decreases with the distance as a function of $||k - k^*||$. We thus have $f_k = f_k(r)$ with $r \equiv ||k - k^*||$. Passing to the continuous domain, the indices in the Gaussian "smoothing filter" $g_{k^*k}$ become continuous variables, and we have $g_{k^*k} \rightarrow g(||k - k^*||, \sigma) \equiv g(r, \sigma)$. Similarly, $f_k(r) \rightarrow f(r)$. Using 2D polar coordinates, the smoothed max-component likelihood $\hat{\mathcal{L}}^\sigma$ becomes a polar integral around the position of the best-matching component: $\hat{\mathcal{L}}^\sigma \sim \int_{\mathbb{R}^2} g(r, \sigma) f(r) dr d\phi$. We are interested in the change of $\hat{\mathcal{L}}^\sigma$ when $\sigma$ undergoes an infinitesimal change. It is trivial to show that for the special case of a constant log-probability profile, i.e., $f(r) = L$, $\mathcal{L}^\sigma$ does not depend on $\sigma$ because Gaussians are normalized, and that the derivative w.r.t. $\sigma$ vanishes:

$$\begin{aligned}
\frac{d\hat{\mathcal{L}}^\sigma}{d\sigma} &\sim \int_0^\infty dr \Big( \frac{r^2}{\sigma^2} - 1 \Big) \exp(-\frac{r^2}{2\sigma^2}) L \\
&= L \int_0^\sigma dr \Big( \frac{r^2}{\sigma^2} - 1 \Big) \exp(-\frac{r^2}{2\sigma^2}) - L \int_\sigma^\infty \Big( \frac{r^2}{\sigma^2} - 1 \Big) \exp(-\frac{r^2}{2\sigma^2}) \\
&\equiv L\mathcal{N} - L\mathcal{P}
\end{aligned} \tag{12}$$

where we have split the integral into parts where the derivative w.r.t. $\sigma$ is negative($\mathcal{N}$) and positive ($\mathcal{P}$). We know that $\mathcal{N} = \mathcal{P}$ since the derivative must be zero for a constant function $f(r) = L$ due to the fact that Gaussians are normalized to the same value regardless of $\sigma$.

For a function $f(r)$ that satisfies $f(r) > L \forall r \in [0, \sigma[$ and $f(r) < L \forall \in ]\sigma, \infty[$, the inner and outer parts of the integral behave as follows:

$$\begin{aligned}
\tilde{\mathcal{N}} &= \int_0^\sigma g(r) \Big( \frac{r^2}{\sigma^2} - 1 \Big) f(r) < \int_0^\sigma g(r) \Big( \frac{r^2}{\sigma^2} - 1 \Big) L = L\mathcal{N} \\
\tilde{\mathcal{P}} &= \int_\sigma^\infty g(r) \Big( \frac{r^2}{\sigma^2} - 1 \Big) f(r) < \int_\sigma^\infty g(r) \Big( \frac{r^2}{\sigma^2} - 1 \Big) L = L\mathcal{P}
\end{aligned} \tag{13}$$

since $f(r)$ is minorized/majorized by $L$ by assumption, and the contributions in both integrals have the same sign for the whole domain of integration. Thus, it is shown that, for $\sigma > 0$

$$\frac{d\hat{\mathcal{L}}^\sigma}{d\sigma} = \tilde{\mathcal{N}} - \tilde{\mathcal{P}} < L\mathcal{N} - L\mathcal{P} = 0 \tag{14}$$

and, furthermore, that this derivative is zero for $\sigma = 0$ because $\hat{\mathcal{L}}^\sigma$ no longer depends on $\sigma$ for this case.

Taking everything into consideration, in a situation where the log-likelihood $\hat{\mathcal{L}}^\sigma$ has reached a stationary point for a given value of $\sigma$, we have shown that:

- The value of $\hat{\mathcal{L}}^\sigma$ depends on $\sigma$.
- Without changing the log-probabilities, we can increase $\hat{\mathcal{L}}^\sigma$ by reducing $\sigma$, assuming that the log-probabilities are mildly decreasing around the BMU.
- Increasing $\hat{\mathcal{L}}^\sigma$ works as long as $\sigma > 0$. At $\sigma = 0$ the derivative becomes $0$.

Thus, $\sigma$ indeed defines an upper bound to $\hat{\mathcal{L}}^\sigma$ which can be increased by decreasing $\sigma$. The assumption of log-probabilities that decrease around the best matching unit (BMU) is reasonable since such a profile maximizes $\hat{\mathcal{L}}^\sigma$. All functions $f(r)$ that, e.g., decrease monotonically around the BMU, fulfill this criterion, where the precise form of the decrease is irrelevant. This proof works identically when not resorting to integrals but using discrete sums.

### A.3 LOG-LIKELIHOOD GRADIENTS

The gradients of $\hat{\mathcal{L}}$ read

$$\frac{\partial \hat{\mathcal{L}}}{\partial \boldsymbol{\mu}_k} = \mathbb{E}_n \left[ \boldsymbol{P}_k \left( \boldsymbol{x}_n - \boldsymbol{\mu}_k \right) \delta_{kk^*} \right]$$

$$\frac{\partial \hat{\mathcal{L}}}{\partial \boldsymbol{P}_k} = \mathbb{E}_n \left[ \left( (\boldsymbol{P}_k)^{-1} - (\boldsymbol{x}_n - \boldsymbol{\mu}_k)(\boldsymbol{x}_n - \boldsymbol{\mu}_k)^T \right) \delta_{kk^*} \right] \tag{15}$$

$$\frac{\partial \hat{\mathcal{L}}}{\partial \pi_k} = \pi_k^{-1} \mathbb{E}_n \left[ \delta_{kk^*} \right].$$

The gradients of $\mathcal{L}$ are obtained by replacing $\delta_{kk^*}$ by the standard GMM responsibilities $\gamma_{nk}$. For the case of a degenerate solution when optimizing $\mathcal{L}$, only a single component $k^*$ has a weight close to $1$, with its centroid and covariance matrix are given by the mean and covariance of the data: $\pi_{k^*} \approx 1$, $\boldsymbol{\mu}_{k^*} = \mathbb{E}[\boldsymbol{X}]$, $\boldsymbol{P}_{k^*}^{-1} = \text{Cov}(\boldsymbol{X})$. In this case, the gradients w.r.t $\boldsymbol{\mu}$ and $\boldsymbol{P}$ vanish. The gradient w.r.t. $\pi_k$ does not vanish, but is $\delta_{kk^*}$ which vanishes after enforcing the normalization constraint.

## A.4 VISUALIZATION OF 2D ANNEALING GRID

In Figure 4, three different states of the $g_k$ are visualized, depending on $\sigma(t)$. Darker pixels indicate larger values. Each $g_k$ is assigned to a single GMM component $k$, which is why the $g_k$ are arranged on the same $\sqrt{K} \times \sqrt{K}$ grid we place the components themselves. An intuitive interpretation of a particular $g_k$ is that it encodes the contribution of neighbouring component log-probabilities on the log-probability of component $k$ entering into max-computation. Over time, $\sigma(t)$ is reduced (middle and right pictures) and thus only component $k$ contributes. Please note that the grid we place the components on is periodic for simplicity, so the $g_k$ are themselves periodic.

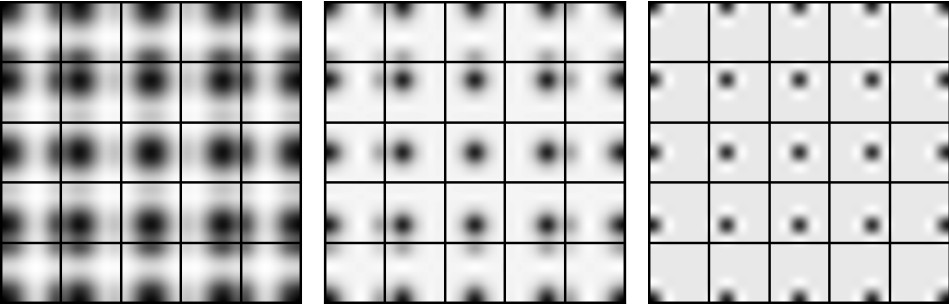

Figure 4: Visualization of different centered Gaussian function $g_k$ states controlled by $\sigma$.

## A.5   Diagrams of Clustering Comparison

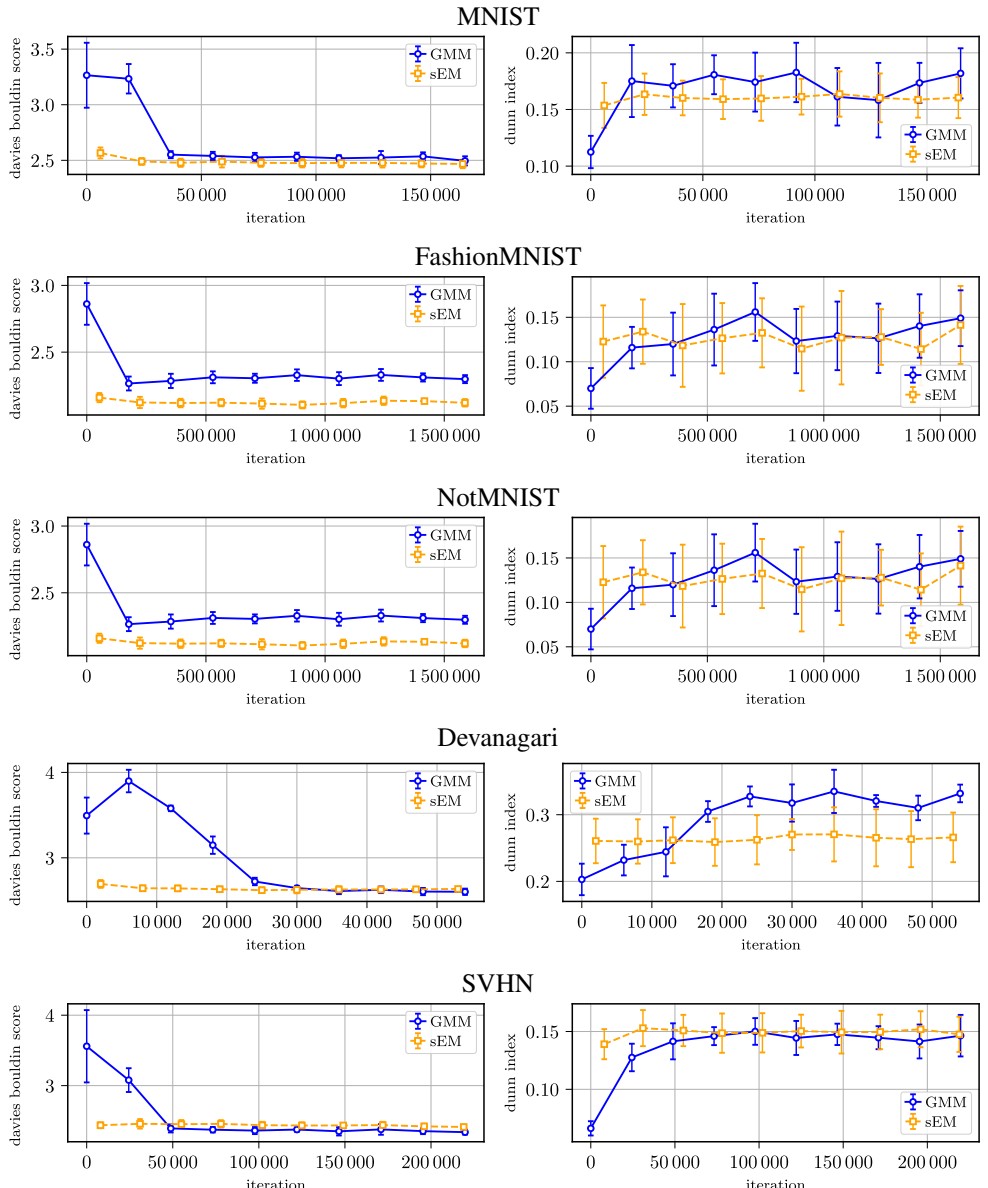

Figure 5: Trend of clustering capabilities for sEM and SGD trained GMM. Comparison by Davies-Bouldin score (smaller is better) and Dunn index (higher is better) for the datasets MNIST, FashionMNIST, NotMNIST, Devanagari and SVHN. The lines visualize the average metric score/index values of 10 repetitions with its standard deviation.

A.6   EFFECT OF THE GMM COMPONENT NUMBER $K$

The number of Gaussian components $K$ is a key parameter for any GMM and has a huge impact on performance. It should be stated clearly that this discussion is not specific any particular fashion of performing GMM *training*, be it by SGD, sEM or EM since all of these methods optimize the log-likelihood. This is why we do not propose a particular way of choosing $K$ for SGD-trained GMMs. For clustering, $K$ can be chosen using standard techniques like BIC or AIC, in addition to priors depending on the data and the concrete application in mind. For density estimation, it is generally assumed that $K$ should be set as high as possible since the real distribution of the data can be approximated in better detail. This set of experiments aims at showing empirically, for completeness, that this "bigger is better" relation holds when training GMMs by SGD. We use the hyper-parameter settings stated in Sec. 4, vary $K$ and record the final log-likelihoods. Results are shown in Tab. 4 and suggest a very clear relationship between $K$ and the (test) log-likelihood obtained at the end of training. We run the experiments for a larger number of epochs to exclude that (non-)convergence effects could play a role here.

Table 4: Log-likelihood values after 50 training epochs using different number of Gaussian components $K \in \{25, 36, 49, 64, 81\}$. Best values of the final log-likelihood are in bold face.

| Components / Dataset | SGD | | | | |
|---|---|---|---|---|---|
| | $K = 5^2$ | $K = 6^2$ | $K = 7^2$ | $K = 8^2$ | $K = 9^2$ |
| | *mean* | *mean* | *mean* | *mean* | *mean* |
| MNIST | 205.4 | 215.2 | 222.7 | 229.6 | **235.1** |
| FashionMNIST | 215.9 | 250.5 | 266.4 | 273.9 | **281.8** |
| Devanagari | $-15.7$ | $-0.7$ | 20.9 | 24.3 | **66.7** |
| Fruits 360 | 12 095.7 | 13 003.0 | 12 953.2 | 12 866.6 | **13193.7** |

