# OpenReview forum: "Gradient-based training of Gaussian Mixture Models for High-Dimensional Streaming Data"
_ICLR.cc/2021/Conference — Reject_

### Official Review · AnonReviewer3 · 2020-10-29
**online gaussian mixture model learning with sgd and smoothed max-component log-likelihood**

**Rating:** 5
**Confidence:** 3

**Review:**

This paper presented a stochastic gradient descent approach to learn a non-stationary high-dimensional Gaussian mixture model from online data. The authors identified 3 challenges - local optima, numerical instability, and catastrophic forgetting, and proposed to address these challenges respective with adaptive annealing, exponential-free approximation, and adaptive SGD learning rate. The proposed approach is demonstrated with several vision/non-vision tasks.

Overall, I feel that the paper is slightly below the borderline. There lacks some theoretical analysis of the proposed ideas, an approach to identify the number of mixture components, and an argument as to why GMM is preferred instead of other representation learning techniques.

Pros:

+ Interesting combination of new research trends (continual learning) and old models (GMM).

Cons:

- Lack of an approach to identify the number of mixture components
- Lack of theoretical justification about the max-component approximation and soft max-component approximation.
- Lack of demonstration on why catastrophic forgetting is avoided and how nonstationary data affects this and other algorithms in experiments.

So the following improvements could improve my ratings:

- An empirical analysis of how the proposed approach avoids catastrophic forgetting with nonstationary data, and a theoretical analysis/comparison between existing approaches, such as regularization, memory replay, and network morning.
- A mathematical justification about the soft max-component approximation: how it is related the classical GMM, to what extent it is an approximation.
- A theoretical or emprical approach to adapt the number of mixture components, for example, through introducing a prior.

---

> ### Author Response · Authors · 2020-11-11
> **Author reply for initiating discussion**
>
> Thank you for the constructive review! As this is an open discussion phase, we would value your feedback to our replies to better understand how we can improve the paper. We will incorporate the obvious improvements right away, and the rest as a function of the feedback in the open discussion phase.
>
> Concerning your remarks:
>
>   4. This is a very fair point, thanks for pointing it out: The experiment 4.4 is purely empirical and the claimed justification for the observed effect (which is plainly there) is not very rigorous. On the other hand, adding, e.g., an EWC term to the loss or performing generative replay would be beyond the scope and space of this paper. As the message of this experiment is not really a key point of the paper, we might simply drop it, and the associated claim, in favor of elaborating more on the key claims (cf. you other suggestions) and treat incremental learning for GMMs in a separate paper. What do you think? --> Update: Ok we went ahead and did it anyway
>
>   5. We have added empirical results (left col in Tab.3) that show that this approximation holds, in all experiments, to an extremely high degree of precision, and a theoretical justification of this fact. As for the approximation itself, we elaborate on p. 3, Sec. 3.2, how this max-component approximation is derived from the "classical" GMM log-likelihood. Is that not exactly what you mean by "how it is related the classical GMM, to what extent it is an approximation"?
>
>   6. We have added a new experiment (A.6) showing that higher K is helpful for density estimation, and a discussion of finding a good K for clustering (discussion section, A.6).
> To respond to your concerns:
>   1) If we use the trained GMMs for clustering, we would use the conventional approaches developed for GMM/k-means to determine an optimal number of components. The fact that we train GMMs by SGD (instead of sEM or EM) should not impact these methods in any way, and thus there is not need for new contributions here (--> p8, discussion of "hyper-parameter selection guidelines")
>   2) When performing density estimation, the number of mixture components follows a "the more the better" principle as elaborated on p. 8 ("Hyper-parameter selection guidelines").

---

### Official Review · AnonReviewer1 · 2020-10-30
**This paper proposes an annealing stochastic gradient descent (SGD) approach for efficiently training Gaussian Mixture Models  on non-stationary, high-dimensional streaming data. Although it is well organized and the idea is acceptable and the traning process is clearly described, I still concern its originality and effectiveness. Moreover, its functions for non-stationary, high-dimensional streaming data are not analyzed and tested deeply.**

**Rating:** 5
**Confidence:** 4

**Review:**

It is clear that efficiectly training Gaussian Mixture Models with SGD on non-stationary, high-dimensional streaming data is very important for practical applications. The aim of this paper is quite good.  In fact, this paper proposes an annealing mechanism for the SGD algorithm and makes the experiments to compare the proposed training algorithm with sEM algorithm on several real-world datasets.  However, I have following major concerns:
(1). The proposed training scheme does not require data-driven parameter initialization (e.g., k-means) , but the data-driven parameter initialization can improve the efficiency. So, I cannot consider this is an advantage of the proposed scheme.
(2).  The  proposed annealing scheme is straightforward，and there is no deep analysis on its performance.
(3).  According to the experimental results, I cannot find out that the proposed training scheme is remarkably better that sEM. By the annealing procedure,  the clustering results should be improved much better.
(4).  The authors claim that the proposed training scheme is  good for non-stationary, high-dimensional streaming data, but there are not much analytic results with the dimenionality and non-stationary streaming data.  The forgetting  results by controlling the annealing parameter is too rough.

---

> ### Author Response · Authors · 2020-11-11
> **Author reply for initiating discussion**
>
> Thank you for the constructive review! As this is an open discussion phase, we would value your feedback to our replies to better understand how we can improve the paper. We will incorporate the obvious improvements right away, and the rest as a function of the feedback in the open discussion phase.
>
> Concerning the points you make:
>
>   (1): In streaming settings, prior, data-driven initialization is impossible because data are not (yet) available. Even if they were, k-means scales very badly for huge high-dimensional datasets so it becomes impractical. And the experiments of Secs. 4.1 and 4.3 clearly show that data-driven and random initializations perform equally well. So the advantage is: we can achieve the same results as with k-means, but in scenarios where k-means is unfeasible or impossible.
>
>   (2): We feel that SGD requires no special mathematical analysis here, and that is probably not what you meant anyway. For the annealing, please refer to the appendix A.2 for an analysis. We show that $\sigma$ always imposes an upper bound on the loss we are optimizing, and that we can improve the loss by decreasing $\sigma$ except when $\sigma \rightarrow 0$, demonstrating that adaptive annealing is stable. We could move this to the main text, would you consider that helpful?
>
>   (3): Thank you for pointing this out, we have stated this more clearly (top of p8).
>   We do NOT wish to show that our SGD can outperform sEM by a large margin, nor do we think this is generally possible. After all, sEM makes use of strong theoretical guarantees that do not hold for SGD. However, SGD can do just as well as sEM, or slightly better, which is surprising given the reasons just mentioned.
>
>   There is one aspect where SGD outperforms sEM by a large margin: for very high-dimensional data like SVHN and Fruits (Tab. 3, discussed at the top of p. 8). Here, sEM does not converge well without using k-means but SGD does and performs much better. We have added another Figure to the main text(Fig.3), plus some text (p7) explaining why this is the case.
>
>   (4): We agree. This experiment is just an "interesting fact" without any theory behind it, and not extremely important for the message of the paper. It might be better to skip this experiment in favor of some more analytical contribution you mentioned earlier. What do you think?  --> Update: in absence of a reply, we did it anyway!

---

### Official Review · AnonReviewer4 · 2020-10-30
**Paper investigates some key research questions in training mixture models (GMMs) but lacks clarity**

**Rating:** 5
**Confidence:** 3

**Review:**

The paper proposes a new approach to train GMMs using SGD under a variety of settings (streaming, concept drift, etc) addressing the issues of catastrophic forgetting, problem of parameter initialization and numerical instability.

These are all important and interesting challenges that can help advance the training of latent variable models in general. However I feel that paper is not presented with sufficient clarity to pass the ICLR bar and the exposition could be greatly improved. It is hard for me to grasp the key findings or takeaways in the paper wrt to other existing baseline methods.

I also have some specific comments:

- The contributions sub-section in 1.3 are vague to read, instead of simply saying - "a novel method", "an automatic annealing procedure" it would be useful to explicitly state what is novel / automatic etc - this would make it easier for the reader to understand the novelty/exact technical contributions made by the authors

- In section 3.2, I am curious how the max-component approximation lower bound compares to the other lower bounds used in EM (e.g. Jensen derived lower bound or evidence lower bound in Variational Inference)? To me the presented lower bound seems like inefficient and loose (e.g. if the values are close in magnitude, then intuitively the gap between the sum and max is going to be very large). Maybe I am mistaken in understanding this bound, but it would need some more explanation and it should be contrasted with the vanilla lower bounds used in EM.

- In section 3.3, the authors mention updating a subset of components to break the symmetry. I am curious how a baseline of simply perturbing the GMM parameters randomly would perform since that would also help break the symmetry?

- I liked the idea of using annealing to avoid local optima and this seems to be one of the key contributions of this work in my opinion. One of my main questions here is: how did the authors measure the value addition of annealing? Did they compare the final solutions obtained by the proposed approach with a baseline (without any annealing) that uses random initialization to deal with local optima?

- Notations in the paper could be improved further to make it more readable. For e.g. in Algorithm 1, the iteration steps are missing in the updates (would be good to use t=1, 2, ..T and use them in the update equations since it is not clear which iterate time steps the parameters on the RHS belong to). Also, maybe I missed this while reading but prec_clipping(..) doesn't seem to be defined near the algorithm section.

- I found some vague statements in the Empirical section which raises lot of follow up questions. Would suggest making the description and analysis of the results more precise. e.g. In the beginning of section 4, authors say: ".. repeated 10 times with identical parameters.." -> what does identical mean? what was varied?

---

> ### Author Response · Authors · 2020-11-11
> **Author reply for initialing discussion**
>
> Thank you for the constructive review! As this is an open discussion phase, we would value your feedback to our replies to better understand how we can improve the paper. We will incorporate the obvious improvements right away, and the rest as a function of the feedback in the open discussion phase.
>
> Answer to the general text("lack of clarity"):
>
>   Could you please elaborate where precisely you do not find our paper sufficiently clear and what we could do about that? Your statement ("However I feel that ...") sounds like a summary judgement of the whole paper, and it is hard to improve the paper based on that. In Sec. 1.3 (contributions), we present a bullet-list of contributions: if we tried to put this more concisely and reduce the number of bullet points, would that make the paper more clear?
>
> Concerning the specific comments:
>   1. Point taken, we rephrased the contributions to that effect. To answer your concerns: Annealing is not novel or remarkable in itself. What is novel is its use in GMM/SGD training which has never been proposed before, and which gives good results.
>
>   2. First point: the max-component approximation is more loose than Jensen (easy to show). Second point: it is still an extremely good approximation for high-dimensional data. We measured that all GMM responsibilities are > 0.99 throughout all experiments, and we qdeed q new experiment to the paper showing these results (--> middle of p.7 and Tab.3)  which justify our approximation. Since this can be expected to hold only for high data dimensions, the paper is restricted to the high-dimensional case (cf. paper title).
>
>   3. Interesting point: we will add a statement to the discussion elaborating on this in the final version (more space). To answer the question here: this might work for the degenerate solutions but not for sparse-component ones. The sparse-component solutions seem to represent very broad basins of attraction, so perturbing the parameters will in general only lead back to sparse-component solutions.
>
>   4. Very good point, thank you: We added a simple experiment showing the value of annealing to the experiments section (--> 4.2). To answer the question: The value of annealing is established by the fact that training never converges when not using annealing (i.e., having a small constant $\sigma$), independently of the used dataset. This can be observed in the prototypes but also from the loss function values which are consistently higher with annealing turned on.
>
>   5. Point taken, notation has been be improved, mainly in Algo. 1
>
>   6. Thanks for pointing this out, we have made this more precise (--> Sec. 4). To answer the question: identical means "identical hyper-parameters but different seeds for random initialization", so as to exclude/reduce the impact of a particular random initialization (similar as for DNN training).

---

### Official Review · AnonReviewer2 · 2020-10-31
**Add annealing feature so that SGD can solve max-component log-likelihood approximation of GMM, but still need more discussion in depth**

**Rating:** 5
**Confidence:** 4

**Review:**

The authors propose a technique to training GMM using SGD instead of (s)EM. The major contributions are:
1. a proposal for numerically stable GMM training by SGD
this is achieved with max-component log-likelihood approximation and an annealed process to smooth the SGD.
2. an automatic annealing procedure that ensures SGD convergence from a wide range of initial conditions without prior knowledge of the data (e.g., no k-means initialization) which is especially beneficial for high-dimensional data
Section 4.4 has shown that such annealing hyper-parameter can control re-learning and retention process, which is good.
However, section 4.1 has shown that with or without random initialization does not impact the perf too much, then why not just use k-means initialization? is it very costly? I believe k-means initialization is also a randomized process.

However, there are some issues:
1. When comparing SGD vs sEM, two strong assumptions are made:
1) SGD annealing has converged and
2) GMM responsibilities are sharply peaked so that a single component has responsibility of around 1
It basically requires the data does not have a lot of noise, where each point can be assigned with to a label with a dominating probability, so what happens if the data has some noise, and how will such solution reacts for different level of noise?
A second question is: is there any theory or bounds to support the convergence assumption? all experiments do SGD for 2 or 3 epochs, how will the loss be like after 3 epochs?
2. No actual examples or discussions are given in terms of numerical instabilities when data dimension is high
3. For section 4.3, the streaming scenario is not clear, is mini-batch size constant? random? if fed data of one sample, is the SGD stilling running for 2 epochs? or halting and wait?

---

> ### Author Response · Authors · 2020-11-11
> **Author Reply  for initiating discussion**
>
> Thank you for the constructive review! As this is an open discussion phase, we would value your feedback to our replies to better understand how we can improve the paper. We will incorporate the obvious improvements right away, and the rest as a function of the feedback in the open discussion phase.
>
> Response to the questions in the text:
>
>   K-means initialization would require having all the data at your disposal which is excluded in streaming settings so we simply cannot do that. And k-means does not scale well to huge high-dimensional datasets which is another point against it in the targeted scenario.
>
> Responses to the mentioned issues:
>
>   1.1 We chose the number of epochs large enough such that annealing always converged in all experiments, for simplicitly. In practice, one would terminate training once a certain annealing radius is reached --> extremely simple scheme, thus of advantage. We added a statement to that effect to the experiments section (--> 4.4)
>
>   1.2 We added experimental results  (--> Tab.3  ) showing that responsibilities are peaked to an extremely high degree (always > 0.99) for all datasets. The reason is the high data dimensionality which leads to large inter-component distances, so the distance to the closest centroid is much larger than the distance to the next-closest one. Since this does not hold for low data dimensions, the paper is restricted to the high-dimensional case (cf. paper title)
>
>   2. Please look on page 3, at the end of Sec. 3.2, we do give a concrete example for numerical instabilies (underflow: a value is taken to be 0 when it is not) which can easily lead to NaNs in later stages. Is that the example you wanted to see?
>
>   3. Please see p. 5, at the very beginning of Sec. 4: we clearly state that the mini-batch size is always set to 1, in all experiments, to closely emulate a streaming setting (cf. paper title).
>   SGD is always applied after a sample is processed, for each sample in 2 epochs. Not sure whether this was what you meant by your question, if not: could you please elaborate more?

---

### Official Review · AnonReviewer5 · 2020-11-10
**The paper proposes a SGD based method to learn GMM in non-stationary and high dimensional setting. The paper is tackling an interesting problem however the contributions of the paper is not clearly supported.**

**Rating:** 5
**Confidence:** 2

**Review:**

A major concern about the paper is related to the unsupported claims and contribution throughout the paper. For example, the way the training copes with distribution shift or alleviate forgetting is not clear or elaborated on. Beyond the abstract and before the empirical validation no theory or justification is provided to substantiate this claim. The idea of the paper and the motivation are very interesting. The experiments look convincing. Writing and presentation are a good start point for improving the paper.

---

> ### Author Response · Authors · 2020-11-10
> **Could you please elaborate?**
>
> Thank you for taking time to perform a review of our paper! In order to start a discussion and/or to start improving our paper, could you please list all of the "unsupported claims and contribution(s)" that you believe we are making "throughout the paper"? And could you please elaborate a bit more on what exactly you dislike about the concept shift experiments? At this detail level, we cannot be sure what exactly it is you are criticizing here, and of course we would like to improve the paper according to your suggestions.
>
> Just a word on concept shift experiments: these are empirical findings.  We train with one set of classes, then retrain with another one, and observe what happens.  It is not a key point of the paper, nor do we claim that it is a huge theoretical breakthrough, it is just an interesting fact.

---

### Decision · Program_Chairs · 2021-01-07
**Final Decision**

**Decision:**

Reject

**Comment:**

This paper proposes training Gaussian mixture models using SGD, creating an algorithm appropriate for streaming data. However, we feel that the current manuscript does not sufficiently support the proposed method, and lacks insight into its workings. The reviewers believed the method lacked justification (while the authors claim to have added theoretical justification to the revised manuscript, I did not see any such new theory), and were not convinced that the method offered a significant improvement on existing methods.